# OpenReview forum: "Benchmarking Vision Language Model Unlearning via Fictitious Facial Identity Dataset"
_ICLR.cc/2025/Conference — ICLR 2025 Poster_

### Official Review · Reviewer_ndqf · 2024-10-25

**Soundness:** 2
**Presentation:** 3
**Contribution:** 2
**Rating:** 5
**Confidence:** 4

**Summary:**

This paper, based on the Right to be Forgotten setting, defines a VLM unlearning scenario that is closer to real-world use cases. The primary contributions are as follows:
1. Formalizing the VLM unlearning tasks. It emphasizes that VLM unlearning algorithms should focus on sensitive information linked to images rather than the visual attributes themselves.
2. Defining a two-stage evaluation pipeline with the Fictitious Facial Identity VQA dataset. In the first stage, personal information is injected into the VLM, and in the second stage, the unlearning algorithm is performed.
3. Providing various metrics for robust evaluation in terms of forget quality and model utility.

**Strengths:**

Based on the "Right to be Forgotten" setting, this paper defines a new VLM unlearning scenario closer to real-world use cases.
The writing is clear.

**Weaknesses:**

1. The rationale behind the research motivation requires further substantiation (Q1).
2. Some experimental details are not clearly described or potentially contain errors. (Q2, Q3, Q6).
3. The analysis of the experimental results does not sufficiently consider the characteristics of unlearning algorithms, namely, the trade-off of model utility and forget quality for unlearning algorithms (Q4).
4. Some metrics appear to lack robustness when the VLM category changes (Q5).

**Questions:**

1. My first question is whether it is necessary to store individual private information within VLMs in real-world use cases. In practical applications, the best approach is to separate individual private information from the VLM and use retrieval-augmented generation (RAG) techniques to provide appropriate responses. Under such techniques, the Right to be Forgotten can be easily ensured by deleting individual private information from the relevant databases. The authors need to further elaborate on the motivation for their research.
2. In line 290, it is described that the score range for GPT-Eval is 0-1, but in Table 1 and Table 2, there are scores that exceed this range. This appears to be an error.
3. Line 366 mentions that “early stopping based on training loss” is employed. Are the results reported in Table 2 based on this early stopping setting? What criteria are used for early stopping? I would like to know more details.
4. Unlearning algorithms involve a trade-off between model utility and forget quality. It might be necessary to compare the forget quality scores at fixed model utility levels, or vice versa. Alternatively, plotting a Model Utility-Forget Quality curve could be more informative. In fact, in Figure 3, representing the effectiveness of the unlearning algorithm with a single point is also unreasonable; a Model Utility-Forget Quality curve would likely be a more appropriate choice.
5. On certain metrics, the VLM category significantly affects the model utility and forget quality of an unlearning algorithm. Why is this the case? Comparing the performance of the unlearning algorithm with the Retain Model reveals many such instances: (1) The difference between the Retain Model and GA for LLaVA-Phi-mini-3B is 42.4 (93.7 v.s. 50.6), whereas, for LLama-3.2-Vision-Instruct-11B, the difference is 84.5 (88.8 v.s. 4.30). (2) The difference between the Retain Model and KL for LLaVA-Phi-mini-3B is -29.8 (12.3 v.s. 42.1), whereas, for LLama-3.2-Vision-Instruct-11B, the difference is -1.5 (12.2 v.s. 13.7). The significant impact of the VLM category on certain metrics raises the question of whether these metrics can provide robust testing results. Please provide a more detailed discussion on this matter.
6. In line 464, it is stated that you "finally decided to fine-tune the parameters from LLM." However, from Table 3, it is evident that the MIA of $E_{x_3}$ is the highest among the four fine-tuning strategies. This choice seems to lack sufficient justification.

---

> ### Author Response · Authors · 2024-11-23
> **Response to Reviewer ndqf25**
>
> Thanks for your comments. We’d like to address your concerns as follows.
>
>
> **Q1: Research motivation for the necessity to store individual private information within VLMs.**
>
> First, we need to clarify that our benchmark is not limited to evaluating how to remove the private information stored within VLMs. Instead, our benchmark focuses on robustly evaluating the effectiveness of unlearning algorithms within the Right to be Forgotten setting. Any image-paired question-answering datasets where VLMs lack prior knowledge can be applied for this evaluation (such as using some fictional creature images).
>
> Additionally, we strongly agree with the approach of separating individual private information from VLMs and using the RAG method to enhance responses. However, in real-world scenarios, since VLMs are trained on massive corpora of data from the web, it is unavoidable that they may memorize and reproduce some sensitive or private data. Therefore, unlearning provides a way to protect private data and remove sensitive information after training.
>
>
>
> **Q2: Score range for GPT-Eval.**
>
>
> We apologize for the lack of clarity in our description of the presented results. We multiplie the GPT scores by 100 to calculate a reasonable average with the other metrics (e.g., Rouge-L, Accuracy, Truth Ration) of model utility. We have also added a corresponding explanation in the line 290-291 of the revised version of our paper.
>
>
> **Q3: Details about the early stopping setting.**
>
> All results in Table 2 are based on the early stopping setting. We provide the hyperparameters used for the unlearning stage in Appendix D.1. For each unlearning algorithm, our initial setup involves fine-tuning the model for 8 epochs. However, for the GA and GD methods, their losses are negative, and as the training steps increase, the loss can become extremely small (even less than -50). When the loss reaches this level (-50), the models completely collapse and do not output any tokens for any question, which is obviously unreasonable. Therefore, we set the early stopping criterion as follows: when the loss exceeds -20, the model stops training. This ensures that the unlearned model remains usable and does not become entirely non-functional. For KL and PO methods, we don't apply the early stopping setting.
>
>
>
> **Q4: Model Utility-Forget Quality curve to compare the forget quality scores at fixed model utility levels.**
>
> Thanks for your constructive comments! We have plotted a Model Utility-Forget Quality curve in Appendix D.4 (Figure 4) in the revised version of our paper. The results reveal that the GD method is the most effective unlearning algorithm when maintaining 60\% of original model performance. Meanwhile, the KL method achieves the highest level of forgetting when retaining 80\% of model performance. However, this method only manages to forget 60\% of face-related private knowledge, highlighting the urgent need for more effective unlearning algorithms tailored to vision-language models (VLMs).
>
>
>
>
> **Q5: Concerns about the robustness of evaluation metrics.**
>
>
> Thank you very much for your detailed observation. The reason for this result lies in the fact that different VLMs exhibit varying degrees of unlearning under the same early stopping settings:
>
> First, it is important to clarify that LLaVA-Phi-mini-3B and LLama-3.2-Vision-Instruct-11B are two completely different VLMs. They have distinct LLMs, scales, vision encoders, and training data. Therefore, we believe that controlling their unlearning degree solely based on a fixed loss threshold (e.g., -20) is overly simplistic. Instead, we suggest that using the model utility-forget quality curve you mentioned could partially address this issue.
>
> Specifically, in Table 2, we can observe that for the GA algorithm, LLama-Vision demonstrates a significantly higher degree of unlearning, which causes its model utility to drop to an almost unusable level. However, LLama-Vision (**-0.2; 93.2 vs. 93.4**) achieves better forget quality compared to LLaVA-Phi (**-0.4; 92.9 vs. 93.3**). Therefore, the results show that LLama-vision has traded lower mod utility for better forget quality, which is reasonable. For the KL algorithm, LLama-Vision also exhibits a significantly higher degree of unlearning —> LLama-Vision: **-1.5 (12.2 vs. 13.7)** > LLaVA-Phi: **-29.8 (12.3 vs. 42.1)**. However, its model utility drops significantly more: LLama-Vision **-32.9 (52.4 vs. 85.3)** compared to LLaVA-Phi **-10.4 (88.6 vs. 78.2)**.
>
> Therefore, we argue that our metrics can provide robust testing results.
>
>
> **Q6: The choice for fine-tuning the parameters from LLM in the ablation study.**
>
> We apologize for the written-error in our paper. In fact, we finally select the Ex4 which finetunes the parameter of the projector and LLM. We have revised it in our edited version.

---

> > ### Author Response · Authors · 2024-11-27
> >
> > Dear Reviewer ndqf25,
> >
> > As the discussion period deadline approaches, we would like to kindly inquire whether we have adequately addressed the concerns raised in your review. If any remaining points require further clarification, please know that we are eager to provide thorough responses to any additional questions or comments you may have.
> >
> > We greatly value your constructive feedback and appreciate your efforts in helping us improve the quality of our manuscript. Thank you once again for your time and consideration.
> >
> > Best regards, The Authors

---

> > ### Comment · Reviewer_ndqf · 2024-11-29
> >
> > Thank you to the author for clarifying some of the questions I raised in my comments.
> >
> > However, I still have the following concerns.
> >
> > For Q1:
> >
> > The authors state in their response that "In real-world scenarios, since VLMs are trained on massive corpora of data from the web, it is unavoidable that they may memorize and reproduce some sensitive or private data. "
> > Is there any reference or experiment that can substantiate this claim? In practice, sensitive data are subject to anonymization before model training, so theoretically, personal information should not be incorporated into the model. In Section 3.1, the experiments shown in Table 1 also indicate that the two MLLMs involved obtained extremely low GPT scores. This suggests that "the VLMs do not possess knowledge of the correct answers prior to fine-tuning." (line 351 in the updated pdf)
> > I am not sure whether existing MLLMs (whether open-source or closed-source) indeed memorize some sensitive or private data.
> >
> > For Q4:
> >
> > 1. In Figure 4, there are instances where a decrease in Forget Quality is accompanied by a decrease in Model Utility, which seems counterintuitive. Could you provide some explanations for this phenomenon?
> > 2. I would like to understand if the following procedure for generating the Model Utility-Forget Quality curve would be more reasonable:
> > (Note: The author's team does not need to conduct new experiments.)
> > Generally, as the unlearning algorithm is executed, Forget Quality should gradually improve while Model Utility tends to decline. One might select appropriate model checkpoints at steps such that Forget Quality is sampled at regular intervals (e.g., recording values of 1, 0.9, 0.8, 0.7, 0.6, 0.5, 0.4, 0.3, 0.2, and 0.1) and then report the Model Utility of the corresponding checkpoint at each Forget Quality score.
> >
> > I cannot determine the sampling rationale for the current Figure 4. The data points on both the horizontal and vertical axes appear to be randomly sampled instead of being evenly spaced.
> >
> > In summary, I believe this is an inspiring work, but it has two significant shortcomings: (1) The motivation is still somewhat inadequately articulated. (2) In the main experiment, the experimental design appears somewhat unreasonable, as it does not account for the common practice of balancing the two objectives (Forget Quality and Model Utility).

---

> ### Author Response · Authors · 2024-11-29
>
> Thank you for your thoughtful question. We would like to address your concerns as follows:
>
> **Q7: References for our motivation**
>
> We agree with your point that "sensitive data are typically anonymized before model training" to reduce the risk of private information being included in the pretraining data. However, to our knowledge, we would like to emphasize that there is no conclusive evidence showing that anonymization techniques can fully eliminate all sensitive or private information from large-scale datasets used for training LLMs or VLMs. Many works have pointed out that the issues of private information memorization and leakage still exist in LLMs [1,2,3,4] and VLMs [5]. As a result, it is necessary to explore additional strategies to ensure the removal of such data. Furthermore, a recent study [6] has shown that VLMs can still make privacy-infringing inferences from previously unseen combinations of text and images. This suggests that even in the absence of explicit private information in the pretraining data, VLMs' powerful inference abilities may still pose privacy risks.
>
> [1] Counterfactual Memorization in Neural Language Models
>
> [2] Quantifying Memorization Across Neural Language Models
>
> [3] Scalable Extraction of Training Data from (Production) Language Models
>
> [4] SHIELD: Evaluation and Defense Strategies for Copyright Compliance in LLM Text Generation
>
> [5] Protecting Privacy in Multimodal Large Language Models with MLLMU-Bench
>
> [6] Private Attribute Inference from Images with Vision-Language Models
>
>
>
> **Q8: Score in Table 1**
>
> We would like to clarify that we created a synthetic dataset consisting of face image + private information pairs to ensure that these sensitive data were not included in the pretraining process, which explains the low GPT scores observed in Table 1. Afterward, we injected these synthetic private data into the VLMs through fine-tuning and then applied unlearning algorithms to remove them. This approach allowed us to fairly evaluate the effectiveness and robustness of various unlearning algorithms, while also highlighting the challenges associated with applying unlearning techniques to VLMs (**this is our main motivation, as mentioned in Q1**). It's important to note that the removal of private information is only one of the potential applications of unlearning methods in VLMs and is not our primary motivation.
>
>
>
> **Q9: Explanation about Figure 4**
>
> Thank you very much for your careful observation. First, we would like to clarify that for each point in Figure 4, we perform uniform sampling by saving checkpoints every 6 steps. We believe the counterintuitive behavior you mention likely comes from the Preference Optimization method, as it shows a "zig-zag" trajectory in the two-dimensional plane. Specifically, in the latter phase of unlearning, model utility does not decrease as forget quality increases; instead, it follows this fluctuating pattern. We think this occurs because this method minimizes the likelihood of ground truth predictions on the forget set, while also accessing the retain set during loss computation, attempting to balance the losses between the two sets. However, this balance is unstable. Similar phenomena appear in other works, such as Figures 6, 26, and 27 in TOFU [7].
>
> Of course, we agree that the approach you suggest, where model utility is evaluated at the same forget quality score, would be an ideal strategy. However, it is difficult to implement. This is because the decline in forget quality is not a uniform process—it may start with a sharp decrease and then level off more gradually. Moreover, the rate of decline differs across different unlearning algorithms. Therefore, even if we evaluate forget quality at each step, it is impossible to obtain regular intervals (e.g., recording values of 1, 0.9, 0.8, 0.7, 0.6, 0.5, 0.4, 0.3, 0.2, and 0.1).
>
> [7] TOFU: A Task of Fictitious Unlearning for LLMs

---

> ### Comment · Reviewer_ndqf · 2024-12-03
>
> Thank you for your detailed rebuttal. I appreciate your efforts in addressing most of my concerns.
> I’ve decided to keep my original rating since further refinement may help this paper achieve the best presentation.

---

### Official Review · Reviewer_cfPb · 2024-11-03

**Soundness:** 3
**Presentation:** 3
**Contribution:** 3
**Rating:** 5
**Confidence:** 5

**Summary:**

This submission touches an important privacy-related topic in vision language model - the forgetting of specified content, e.g. unlearning. To evaluate the performance of unlearning, the authors construct a Facial Identity Unlearning Benchmark (FIUBENCH), with protocol and several methods as baseline. This is an interesting work. From the reviewer's point of view, this is still a preliminary work in considering of dataset size, and the methods of evaluation and face generation. However, it is worthwhile to continue to advance it to become a consensus for the community.

**Strengths:**

A novel dataset, Facial Identity Unlearning Benchmark (FIUBENCH) is constructed which can support the evaluation of the ‘Right to be Forgotten’ in Visual Language Model. Possible privacy risks are avoided through fictitious facial images. This submission is well written and easy to follow. The protocol provides settings to support different kinds of evaluations, including membership inference attack and adversarial privacy attack, etc.

**Weaknesses:**

The database size is too small to fit the task of evaluate the performance of unlearning in the wild. Although the fictitious facial images avoid the privacy risk, the synthetic images bring some flaws, such as artifacts in style, and these could become an unexpected feature for recognition. In addition the database has taken some action to ‘Filtering out similar faces with K-means’, which leads to an imposed environment for face recognition, and make the evaluation is far from real world case.

**Questions:**

As mentioned in weakness.

---

> ### Author Response · Authors · 2024-11-23
> **Response to Reviewer cfPb03**
>
> Thanks for your comments. We’d like to address your concerns as follows.
>
> **Q1: Database size too small.**
>
> Our dataset pairs each face with 20 QA questions, resulting in 8,000 VLM question-answer pairs. This amount of data is sufficient for evaluating the unlearning problem. In comparison, the previous TOFU unlearning benchmark for LLMs [1] involves only 200 authors, highlighting that our dataset offers significantly more data for analysis.
>
>
> **Q2: Evaluation is far from the real-world case (artifacts in style and imposed K-means filtering).**
>
>
> The purpose of our paper is not to evaluate the potential privacy issues in the real world. Instead, our benchmark aims to simulate the Right to be Forgotten setting and robustly evaluate the effectiveness of different unlearning algorithms. In fact, any image-paired question-answering datasets where VLMs lack prior knowledge can be applied for this evaluation. For easy understanding of VLM unlearning, we still choose to use the synthetic face dataset.
>
> Due to the ethical, legal, and privacy concerns of regulatory constraints by IRB, we do not use datasets with real faces. However, we still demonstrate that our synthetic faces are similar to real ones by showing that they have comparably low distances.
>
> Here we randomly selected 5000 synthetic，celebrity's, private facial images from the SFHQ dataset, the CelebA dataset [1] and the human faces dataset [2]. Since most celebrity's faces are enhanced by makeup and other alterations, they may not accurately represent the distribution of real-world individual faces. Therefore, we also included the human faces dataset, a comprehensive facial dataset covering diverse creeds, races, and age groups. We choose to measure the distribution distances between different face sets using Fréchet Inception Distance (FID) [3] with CLIP features, which are commonly used by VLMs to represent the visual features of images. The results are as follows:
>
>
> | CLIP      |           |         | |
> |-----------|-----------|---------|-----------|
> |           | Synthetic | Private | Celebrity |
> | Synthetic | 0.00      | 19.71   | 29.86     |
> | Private   | 19.71     | 0.00    | 27.51     |
> | Celebrity | 29.86     | 27.51   | 0.00      |
>
>
> Our observations reveal that the distribution difference (FID) between Synthetic and Private faces is smaller than that between Private and Celebrity faces when measured using CLIP features. This indicates that, for VLMs, the difference between synthetic faces and real faces is even smaller than the variability within the distribution of real faces.
>
>
> We use K-means filtering for two main reasons:
>
> (1) To ensure the diversity of synthetic faces. Since people have all kinds of faces in the real life, we aim to closely approximate real-world scenarios that we used the K-means filtering method to ensure that these 400 synthetic face images include a diverse range of facial types. This approach demonstrates that VLMs can retain a variety of facial features along with their corresponding private information (shown in Table 1).
>
> (2) Reducing similar images improves finetuning efficiency.We present in the table below the impact of using randomly selected synthetic face images versus synthetic face images obtained through K-means filtering on the learning stage.
>
> |                  | Kmeans  | Kmeans |  Random |   Random   |
> |------------------|---------|--------|---------|---------|
> |                  | Rouge-L | GPT    | Rouge-L | GPT   |
> | LLaVA-Phi-3-mini | 93.30   | 85.80  | 83.79   | 73.89 |
>
>
> The results show that, under the same condition of fine-tuning for 10 epochs, using more diverse synthetic face images enables VLMs to memorize relevant information more accurately.
>
>
>
>
> [1] Maini, Pratyush, Zhili Feng, Avi Schwarzschild, Zachary Chase Lipton, and J. Zico Kolter. TOFU: A Task of Fictitious Unlearning for LLMs. In ICLR 2024 Workshop on Navigating and Addressing Data Problems for Foundation Models.

---

> > ### Author Response · Authors · 2024-11-27
> >
> > Dear Reviewer cfPb03,
> >
> > As the discussion period deadline approaches, we would like to kindly inquire whether we have adequately addressed the concerns raised in your review. If any remaining points require further clarification, please know that we are eager to provide thorough responses to any additional questions or comments you may have.
> >
> > We greatly value your constructive feedback and appreciate your efforts in helping us improve the quality of our manuscript. Thank you once again for your time and consideration.
> >
> > Best regards, The Authors

---

### Official Review · Reviewer_3WKf · 2024-11-04

**Soundness:** 3
**Presentation:** 4
**Contribution:** 3
**Rating:** 6
**Confidence:** 5

**Summary:**

This paper introduces FIUBENCH, a benchmark designed to evaluate unlearning algorithms for Vision Language Models (VLMs) under the Right to be Forgotten setting. FIUBENCH includes a Fictitious Facial Identity VQA dataset and a two-stage evaluation pipeline to control information exposure levels. To handle VLMs’ ability to process semantically similar queries, FIUBENCH incorporates robust evaluation metrics, including membership inference and adversarial privacy attacks. Initial results on four baseline algorithms show limitations in unlearning performance, with trade-offs between model utility and forget accuracy.

**Strengths:**

This paper systematically examines forgetting in Vision Language Models and introduces FIUBENCH, a new benchmark for robust evaluation of unlearning algorithms.
The paper is well-written and easy to understand.

**Weaknesses:**

The proposed benchmark uses a forget set and a retain set to assess the forgetting quality and model utility of unlearning algorithms. However, is this setting appropriate? In my view, the privacy concerns in Vision Language Models are more about forgetting specific sensitive information, such as identity or email, rather than simply forgetting individual samples.
The forget set is limited to 5% of the total dataset, comprising only 20 images. Could you explain the rationale behind selecting this specific proportion? How was this number determined?

**Questions:**

Please refer to the Strengths and Weaknesses.

---

> ### Author Response · Authors · 2024-11-23
> **Response to Reviewer 3WKf04**
>
> Thanks for your comments. We’d like to address your concerns as follows.
>
> **Q1: Concerns about the appropriateness of the setting with forget set and retain set.**
>
> This setting is appropriate and it is commonly-used setting for unlearning, referring to previous work [1, 2]. As outlined in Section 2.3, after fine-tuning the VLMs on the entire dataset, which includes both the retain and forget sets, we apply the unlearning algorithm exclusively to the forget set. This setting effectively evaluates whether the model can precisely forget the sensitive information in the forget set while preserving the utility derived from the personal information in the retain set.
>
> Additionally, our benchmark does not aim to directly forget individual samples. Instead, it focuses on forgetting question-answer pairs that explicitly contain specific sensitive information, such as medical conditions and hospitalization records, as illustrated in Figure 1.
>
> **Q2: Rationale for selecting the 5% forgetting proportion.**
>
>
> Our paper introduces a range of forgetting difficulties, from easy to hard, based on the forgetting proportion. We present a 5% forgetting proportion in our main table as it represents a moderate setting. As shown in Figure 3, we also provide results for 1% and 10% forgetting proportions, representing easier and harder settings, respectively. Besides, these three forgetting proportions are also common practices in previous work [1, 2].
>
>
> [1] Maini, Pratyush, Zhili Feng, Avi Schwarzschild, Zachary Chase Lipton, and J. Zico Kolter. TOFU: A Task of Fictitious Unlearning for LLMs. In ICLR 2024 Workshop on Navigating and Addressing Data Problems for Foundation Models.
>
> [2] Protecting Privacy in Multimodal Large Language Models with MLLMU-Bench

---

### Official Review · Reviewer_9p6D · 2024-11-05

**Soundness:** 2
**Presentation:** 3
**Contribution:** 2
**Rating:** 5
**Confidence:** 4

**Summary:**

This paper introduces Facial Identity Unlearning Benchmark (FIUBENCH), a VLM unlearning benchmark designed to robustly evaluate the effectiveness of unlearning algorithms under the Right to be Forgotten setting. Moreover, FIUBENCH further incorporates membership inference attacks and adversarial privacy extraction to robustly evaluate unlearning performance, testing whether the private information is unlearned even under attacks.

**Strengths:**

1.	Unlike unlearning in LLMs, which primarily focuses on forgetting sensitive text information, unlearning in VLMs extends to both images and text. This paper formalizes VLM unlearning as the task of unlearning private image and text-paired information.
2.	To study privacy under the Right to be Forgotten scenario, a two-stage evaluation pipeline with Fictitious Facial Identity VQA dataset is proposed.

**Weaknesses:**

1.	This paper proposes FIUBENCH, a VLM unlearning benchmark designed to robustly evaluate the effectiveness of unlearning algorithms under the Right to be Forgotten setting, which is interesting. However, the effectiveness of this benchmark is unclear.
2.	Since the faces are generated by StyleGAN2, it is necessary to evaluate the distance between the generated face distribution and the real one. From the figure 1, the synthetic face images seem different from the real faces. Will it hurt the evaluations on the Vision Language models.
3.	For the experiments, you’d better involve more Vision Language models for evaluations.

**Questions:**

My major concerns lie in the effectiveness of the proposed benchmark and the experiments. If you can well address these problems, I am happy to improve my rating.

---

> ### Author Response · Authors · 2024-11-24
> **Response to Reviewer 9p6D04**
>
> Thanks for your comments. We’d like to address your concerns as follows.
>
> **Q1: Effectiveness of the benchmark.**
>
> We demonstrate the effectiveness of our benchmark through the following points:
>
> Our benchmark effectively establishes a VLM unlearning environment aligned with the 'Right to Be Forgotten' framework. As demonstrated in Section 3.2, it achieves this with minimal prior knowledge of the pre-trained model, evidenced by the GPT scores below 0.01. Additionally, the high GPT score exceeding 80 after fine-tuning indicates that the VLMs successfully acquire fictitious knowledge during the initial learning phase. These findings confirm that our benchmark accurately simulates the 'Right to Be Forgotten' scenario by precisely controlling the source of information and the exposure levels of the dataset's knowledge prior to unlearning, particularly for rarely occurring personal information.
>
> Additionally, our benchmark evaluates several baseline unlearning methods for VLM unlearning, revealing significant trade-offs between forgetting quality and model utility in current baseline algorithms. It also shows that the alignment-based method (Preference Optimization) fails to effectively remove knowledge from VLMs when subjected to our robust evaluation framework.
>
> To further demonstrate the effectiveness of our benchmark, we also plotted a Model Utility-Forget Quality curve in Appendix D.4 (Figure 4) in the revised version of our paper. The results reveal that the GD method is the most effective unlearning algorithm when maintaining 60% of original model performance. Meanwhile, the KL method achieves the highest level of forgetting when retaining 80% of model performance. However, this method only manages to forget 60% of face-related private knowledge, highlighting the urgent need for more effective unlearning algorithms tailored to vision-language models (VLMs).
>
> **Q2: Concerns about synthetic faces hurting the evaluation.**
>
> To ensure that synthetic faces would not hert the evaluation, we calculate the distribution distance between synthetic faces and real faces. Here we randomly selected 5000 synthetic，celebrity's, private facial images from the SFHQ dataset, the CelebA dataset [1] and the human faces dataset [2]. Since most celebrity's faces are enhanced by makeup and other alterations, they may not accurately represent the distribution of real-world individual faces. Therefore, we also included the human faces dataset, a comprehensive facial dataset covering diverse creeds, races, and age groups. We choose to measure the distribution distances between different face sets using Fréchet Inception Distance (FID) [3] with CLIP features, which are commonly used by VLMs to represent the visual features of images. The results are as follows:
>
>
> | CLIP      |           |         | |
> |-----------|-----------|---------|-----------|
> |           | Synthetic | Private | Celebrity |
> | Synthetic | 0.00      | 19.71   | 29.86     |
> | Private   | 19.71     | 0.00    | 27.51     |
> | Celebrity | 29.86     | 27.51   | 0.00      |
>
>
> Our observations reveal that the distribution difference (FID) between Synthetic and Private faces is smaller than that between Private and Celebrity faces when measured using CLIP features. This indicates that, for VLMs, the difference between synthetic faces and real faces is even smaller than the variability within the distribution of real faces.
>
> Finally, we also want to point out that our benchmark focuses on robustly evaluating the effectiveness of unlearning algorithms within the Right to be Forgotten setting. Any image-paired question-answering datasets where VLMs lack prior knowledge can be applied for this evaluation.

---

> ### Author Response · Authors · 2024-11-24
> **Response to Reviewer 9p6D04**
>
> **Q3: More VLMs for evaluations.**
>
>
> To further illustrate the effectiveness of our benchmark, we incorporate Idefics2 [4] for evaluations. Idefics2 is an open multimodal model that accepts arbitrary sequences of image and text inputs and produces text outputs. Its underlying LLM is Mistral-7B-v0.1. In total, we tested three VLMs, each differing in size and LLM architecture (Mistral-7B-v0.1, LLama-3.2-Vision-Instruct-11B, and LLaVA-Phi-mini-3B). The results for Idefics2 are summarized as follows:
>
>
> | Method | Model Utility |       |             | Forget Quality |              |        |
> |--------|---------------|-------|-------------|----------------|--------------|--------|
> |        | Rouge-L       | GPT   | Truth Ratio | KS-Test        | Exact Match | MIA |
> | Retain | 88.67         | 82.90 | 77.33       | 0.00           | 12.52        | 11.55  |
> | GA     | 3.88          | 0.32  | 40.62       | **-0.54**          | 1.89         | **14.46**  |
> | GD     | 22.20         | 8.40  | 57.51       | -1.40          | 5.20         | 16.26  |
> | KL     | **75.48**         | 38.00 | **65.89**       | -12.86         | 28.43        | 68.03  |
> | PO     | 64.01         | **40.77** | 62.88       | -5.81          | **0.28**         | 52.76  |
>
>
>
> The results indicate that methods like GA and GD, which maximize the likelihood of ground truth predictions, exhibit significantly better forgetting quality. However, this comes at the cost of a substantial drop in model utility. This is because, during the unlearning process, the loss continues to decrease without bounds (sometimes even dropping below -50). Although we applied an early stopping strategy (stopping training when the loss falls below -20), over-unlearning still occurs (see our responses to reviewer ndqf25: Q3 and Q5 for more details). On the other hand, the KL method, constrained by the Kullback-Leibler loss, avoids over-unlearning. Similar observations apply to LLaVA-Phi and LLama-Vision.
>
> To better compare the forgetting quality of different unlearning methods at the same level of model utility, we included a model utility-forget quality curve figure in Appendix D.4 of the revised version of our paper. We believe you will find more of the details you’re looking for there.
>
>
> [1] Large-scale CelebFaces Attributes (CelebA) Dataset
>
> [2] https://www.kaggle.com/datasets/ashwingupta3012/human-faces
>
> [3]  GANs Trained by a Two Time-Scale Update Rule Converge to a Local Nash Equilibrium
>
> [4] https://huggingface.co/HuggingFaceM4/idefics2-8b

---

> > ### Author Response · Authors · 2024-11-27
> >
> > Dear Reviewer 9p6D04,
> >
> > As the discussion period deadline approaches, we would like to kindly inquire whether we have adequately addressed the concerns raised in your review. If any remaining points require further clarification, please know that we are eager to provide thorough responses to any additional questions or comments you may have.
> >
> > We greatly value your constructive feedback and appreciate your efforts in helping us improve the quality of our manuscript. Thank you once again for your time and consideration.
> >
> > Best regards,
> > The Authors

---

> > > ### Comment · Reviewer_9p6D · 2024-12-03
> > >
> > > Thank you for your detailed response. Some of my concerns have been addressed, but I still have a few questions regarding Q2. Since the generated images differ significantly from real faces, why is the Fréchet Inception Distance (FID) between the synthetic and private faces smaller than the FID between private and celebrity faces when measured using CLIP? Additionally, for Q3, given the availability of many recent vision-language models, it would be beneficial to include a broader range of models in the evaluation and provide a more in-depth analysis. Overall, I maintain my original rating.

---

> > > > ### Author Response · Authors · 2024-12-03
> > > >
> > > > **Q4: why is the Fréchet Inception Distance (FID) between the synthetic and private faces smaller than the FID between private and celebrity faces when measured using CLIP**
> > > >
> > > >
> > > > We argue that since the FID metric essentially calculates the Fréchet distance between two feature distributions (Gaussian distributions), the result showing a smaller FID between synthetic and private faces compared to the FID between private and celebrity faces suggests that, in the features extracted by CLIP, synthetic faces are closer to private faces. This could be because celebrity faces are mostly of middle-aged or young adults, often with heavy makeup, which causes more noticeable feature differences in CLIP.  It is notable that almost all VLMs use CLIP features to generate visual tokens. Furthermore, we also tested using features from Inception V3 to compute the FID metric, with the following results:
> > > >
> > > > | Inception V3 |           |         |           |
> > > > |-----------|-----------|---------|-----------|
> > > > |           | Synthetic | Private | Celebrity |
> > > > | Synthetic | 0.00      | 51.55   | 52.82     |
> > > > | Private   | 51.55     | 0.00    | 49.46     |
> > > > | Celebrity | 52.82     | 49.46   | 0.00      |
> > > >
> > > >
> > > > The results show that there is little difference in the visual features extracted by Inception V3 among the three groups. This suggests that using synthetic faces can be applicable to real-world scenarios. In fact, many face recognition systems now incorporate synthetic faces as training data to improve performance [1,2,3].
> > > >
> > > > [1] IDiff-Face: Synthetic-based Face Recognition through Fizzy
> > > > Identity-Conditioned Diffusion Models
> > > >
> > > > [2] Digi2Real: Bridging the Realism Gap in Synthetic Data Face Recognition via Foundation Models
> > > >
> > > > [3] AnyFace++: A Unified Framework for Free-style Text-to-Face Synthesis and Manipulation
> > > >
> > > >
> > > > **Q5: include a broader range of models in the evaluation**
> > > >
> > > > We would like to clarify that the motivation of our paper is to fairly evaluate the effectiveness and robustness of various unlearning algorithms, while also highlighting the challenges associated with applying unlearning techniques to VLMs. Therefore, what we should consider is testing different unlearning algorithms, rather than testing many VLMs. In this field, many influential previous works have only tested two models (VLMs or LLMs) while we have tested 3 VLMs with different LLMs. For example, TOFU tested Llama-2-7B and Phi-1.5B [4]; Textual Unlearning tested LLaVA-1.5-7B and LLaVA-v1.6-7B [5]; MLLMU-Bench tested LLaVA-1.5-7B and Idefics-2-8B [6]; LLM unlearning tested OPT-1.3B, OPT-7B, and Llama-2-7B [7]. Based on this, we believe that the reviewer's suggestion is not reasonable.
> > > >
> > > > [4] TOFU: A Task of Fictitious Unlearning for LLMs (COLM 2024)
> > > >
> > > > [5] Can Textual Unlearning Solve Cross-Modality Safety Alignment? (EMNLP 2024)
> > > >
> > > > [6] Protecting Privacy in Multimodal Large Language Models with MLLMU-Bench
> > > >
> > > > [7] Large Language Model Unlearning (ICLR 2024)

---

### Official Review · Reviewer_1Bmo · 2024-11-05

**Soundness:** 3
**Presentation:** 3
**Contribution:** 3
**Rating:** 6
**Confidence:** 3

**Summary:**

This paper introduces an unlearning benchmark for Vision Language Models (VLMs) under the Right to be Forgotten setting. After defining the VLM unlearning tasks, this benchmark assigns a two-stage evaluation pipeline with a newly proposed Fictitious Facial Identity VQA dataset. The proposed benchmark offers a comprehensive evaluation by computing both forget quality and model utility, with further assessment under membership inference attack and adversarial privacy extraction. Another contribution of the work is its evaluating four baseline unlearning algorithms, which indicates that none of them achieve good unlearning performance considering both model utility and forget quality. In addition, the divergent performance of Preference Optimization with and without membership inference attacks underscores the importance of privacy attacks for robust evaluations. This benchmark is good to foster the community’s further research on developing better unlearning methods for VLMs under the setting of Right to be Forgotten.

**Strengths:**

1. This paper addresses an important ethics problem of AI, i.e., to fulfill the right to be forgotten for VLM, which is underexplored relatively. To my understanding, few works have been done in the literature.

2. The benchmark, together with the evaluation metrics, is validated, especially via the assessment of four baseline unlearning algorithms. The results imply that existing unlearning algorithms are far from being mature when considering both model utility and forget quality.

3. The benchmark is good to foster the community’s further research on developing better unlearning methods for VLMs.

**Weaknesses:**

I am confused by the dataset constructed. As described in Line 165~172, 400 faces are sampled, which are then divided into 400 clusters by using K-means algorithm. How can 400 faces clustered into 400 clusters?

And, to me, a dataset with 400 faces is relatively small for evaluating the unlearning problem. I am also not convinced why only synthetic faces are used for this evaluation. Is there any difference between real faces and synthetic faces for this evaluation purpose?

**Questions:**

Please reply my concerns mentioned in the Weaknesses part.

---

> ### Author Response · Authors · 2024-11-23
> **Response to Reviewer 1Bmo04**
>
> Thanks for your comments. We’d like to address your concerns as follows.
>
> **Q1: Details and purpose of K-means algorithm in dataset construction.**
>
> We apologize for any confusion caused by the statement regarding “400 sampled faces” in lines 165–172. We clarify that the face images are sampled from Part 4 of the SFHQ dataset, which consists of 125,754 high-quality 1024x1024 curated face images. These images were generated using "inspiration" images sampled from the Stable Diffusion v2.1 text-to-image generator with various face portrait prompts. Therefore, we clustered these 125,754 images into 400 clusters and selected the central face of each cluster to construct our benchmark. This approach ensures diversity among the evaluation candidate faces in our dataset and simplifies the face feature learning process. The details of the K-means clustering process have been included in Appendix C of the revised version of the paper.
>
>
> **Q2: Concerns about small datasets and the usage of synthetic faces instead of real faces.**
>
>
> Our dataset pairs each face with 20 QA questions, resulting in 8,000 VLM question-answer pairs. This amount of data is sufficient for evaluating the unlearning problem. In comparison, the previous TOFU unlearning benchmark for LLMs [1] involves only 200 authors, highlighting that our dataset offers significantly more data for analysis.
>
> According to our benchmark construction process, we create a profile for each facial image and generate 20 corresponding QA pairs based on these profiles. These profiles may include sensitive information such as health records and criminal histories. If real facial images were used, associating them with their actual information (e.g., health records) would undoubtedly lead to personal information leaks. On the other hand, if fictitious information were created for these real facial images, it could be mistakenly associated with actual individuals, leading to misunderstandings and stigmatization. Moreover, such associations could violate privacy protection regulations (e.g., IRB) and raise ethical concerns. Therefore, we chose to use synthetic facial images.
>
>
>
> To calculate the distribution difference between synthetic faces and real faces, we randomly selected 5000 synthetic，celebrity's, private facial images from the SFHQ dataset, the CelebA dataset [2] and the human faces dataset [3]. Since most celebrity's faces are enhanced by makeup and other alterations, they may not accurately represent the distribution of real-world individual faces. Therefore, we also included the human faces dataset, a comprehensive facial dataset covering diverse creeds, races, and age groups. We choose to measure the distribution differences between different face sets using Fréchet Inception Distance (FID) [4] with CLIP features. The results are as follows:
>
>
>
> | CLIP      |           |         | |
> |-----------|-----------|---------|-----------|
> |           |  **Synthetic** |  **Private** |  **Celebrity** |
> | **Synthetic** | 0.00      | 19.71   | 29.86     |
> | **Private**  | 19.71     | 0.00    | 27.51     |
> | **Celebrity** | 29.86     | 27.51   | 0.00      |
>
>
> CLIP features, commonly used by VLMs, represent the visual features of images. Our observations reveal that the distribution difference (FID) between Synthetic and Private faces is smaller than that between Private and Celebrity faces when measured using CLIP features. This indicates that, for VLMs, the difference between synthetic faces and real faces is even smaller than the variability within the distribution of real faces.
>
>
> Finally, we also want to point out that our benchmark focuses on robustly evaluating the effectiveness of unlearning algorithms within the Right to be Forgotten setting. Any image-paired question-answering datasets where VLMs lack prior knowledge can be applied for this evaluation.
>
> [1] Maini, Pratyush, Zhili Feng, Avi Schwarzschild, Zachary Chase Lipton, and J. Zico Kolter. TOFU: A Task of Fictitious Unlearning for LLMs. In ICLR 2024 Workshop on Navigating and Addressing Data Problems for Foundation Models.
>
> [2] Large-scale CelebFaces Attributes (CelebA) Dataset
>
> [3] https://www.kaggle.com/datasets/ashwingupta3012/human-faces
>
> [4]  GANs Trained by a Two Time-Scale Update Rule Converge to a Local Nash Equilibrium

---

### Meta-Review · Area_Chair_6DmT · 2024-12-17

**Metareview:**

The paper introduces FIUBENCH, a benchmark for evaluating unlearning algorithms in Vision-Language Models (VLMs) under the "Right to be Forgotten" setting, using synthetic facial data and a two-stage evaluation pipeline. There are also some weaknesses of this paper, including the small dataset size, reliance on synthetic faces with potential artifacts, and unclear rationale for some experimental design choices. Despite these limitations, the paper’s novelty and relevance to a growing privacy challenge justify its acceptance, as it lays a foundation for future research in VLM unlearning.

**Additional Comments On Reviewer Discussion:**

Reviewers raised concerns about the dataset's small size, reliance on synthetic faces, unclear clustering methodology, and robustness of evaluation metrics. The authors clarified the use of synthetic data to address ethical concerns, demonstrated its distribution similarity to real faces via FID scores, and justified the dataset size by referencing comparable benchmarks. They provided additional experimental results with more models, detailed evaluation settings, and introduced a utility-forget quality curve for better clarity. While some concerns about motivation and experimental design remained, the authors’ thorough responses and added analyses strengthened the paper's validity, justifying its acceptance.

---

### Decision · Program_Chairs · 2025-01-22

Accept (Poster)